# Andreev molecules in semiconductor nanowire double quantum dots

Zhaoen Su[1], Alexandre B. Tacla[2], Moïra Hocevar[3,4], Diana Car[5], Sébastien R. Plissard [6],
Erik P.A.M. Bakkers [5,7], Andrew J. Daley[2], David Pekker[1] & Sergey M. Frolov[1]

Chains of quantum dots coupled to superconductors are promising for the realization of the Kitaev model of a topological superconductor. While individual superconducting quantum dots have been explored, control of longer chains requires understanding of interdot coupling. Here, double quantum dots are defined by gate voltages in indium antimonide nanowires. High transparency superconducting niobium titanium nitride contacts are made to each of the dots in order to induce superconductivity, as well as probe electron transport. Andreev bound states induced on each of dots hybridize to define Andreev molecular states. The evolution of these states is studied as a function of charge parity on the dots, and in magnetic field. The experiments are found in agreement with a numerical model.

[1] Department of Physics and Astronomy, University of Pittsburgh, Pittsburgh, PA 15260, USA. [2] Department of Physics and SUPA, University of Strathclyde, Glasgow G4 0NG, UK. [3] Universite Grenoble Alpes, F-38000 Grenoble, France. [4] CNRS, Institut Neel, F-38000 Grenoble, France. [5] Department of Applied Physics, Eindhoven University of Technology, 5600 MB Eindhoven, The Netherlands. [6] LAAS CNRS, Université de Toulouse, 31031 Toulouse, France. [7] QuTech and Kavli Institute of Nanoscience, Delft University of Technology, 2628 CJ Delft, The Netherlands. Correspondence and requests for materials should be addressed to S.M.F. (email: frolovsm@pitt.edu)

Quantum simulation is a way to study unexplored Hamiltonians by mapping them onto assemblies of well-understood quantum systems[1] such as ultracold atoms in optical lattices[2], trapped ions[3] or superconducting circuits[4]. Semiconductor nanostructures which form the backbone of classical computing hold largely untapped potential for quantum simulation[5–7]. In particular, chains of quantum dots in semiconductor nanowires can be used to emulate the ground states of one-dimensional Hamiltonians such as the toy model of a topological p-wave superconductor[8–11]. In this case semiconductor quantum dots need to be coupled to superconducting reservoirs, a coupling which induces Andreev bound states. These states are well established for single quantum dots[12–18].

Here, we realize a building block of a p-wave chain model, a double quantum dot with niobium titanium nitride superconducting contacts, in an indium antimonide nanowire[19]. In each dot, tunnel coupling to a superconductor induces Andreev bound states. We demonstrate that these states hybridize to form the double dot Andreev molecular states. We establish the parity and the spin structure of Andreev molecular levels by monitoring their evolution in electrostatic potential and magnetic field. Understanding Andreev molecules is a step toward building longer chains which are predicted to generate Majorana bound states at the end sites[20, 21]. Two superconducting quantum dots are already sufficient to test the fusion rules of Majorana bound states, a milestone towards fault-tolerant topological quantum computing[22–25].

## Results

**Devices**. In order to realize Andreev molecules, we fabricate a device depicted in Fig. 1a. Superconductivity in the InSb

nanowire is induced by two NbTiN contacts placed on top of the nanowire[26], the segments of the wire below the contacts labeled $S_L$ and $S_R$ act as superconducting reservoirs for the left and right dots. The reservoirs are characterized by the induced gap $\Delta \sim 400$ µeV. We use voltages on five electrostatic gate electrodes placed under the nanowire to define the two quantum dots. Voltages on the two outer gates set the couplings $\Gamma_L$ and $\Gamma_R$ to the superconducting reservoirs. Gate voltages $V_L$ and $V_R$ control the chemical potentials on the left and right dots. The middle gate labeled $V_t$ controls the coupling $t$ between the dots. While all couplings are tunable in a wide range, here we focus on the regime where the system is approximately left/right symmetric, and with $\Gamma_L$, $\Gamma_R > t$. In this regime the two dots are strongly coupled to their respective superconducting reservoirs and weakly coupled to each other. The charging energy on each dot $U \sim 1 - 2$ meV $> \Delta$ thus the dots can be filled by electrons one at a time rather than in Cooper pairs.

**Andeev molecules**. In superconductor-semiconductor hybrid structures, electrons arriving from a semiconductor with energies below the superconducting gap are prohibited from entering the superconductor and are reflected back into the semiconductor as quasiholes via Andreev reflection[27]. Through this mechanism, an electron-hole standing wave, known as an Andreev bound state, can form in the semiconductor. In a single quantum dot, Andreev bound state spectrum consists of a spin-singlet state (S) which is a superposition of 0 and 2 electrons on the quantum dot, and two doublet states $D_\uparrow$ and $D_\downarrow$, both of which correspond to a single electron on a quantum dot either in the spin up or spin down state. In Fig. 1b, we depict the Andreev spectra of two decoupled quantum dots along the energy level detuning axis, meaning that

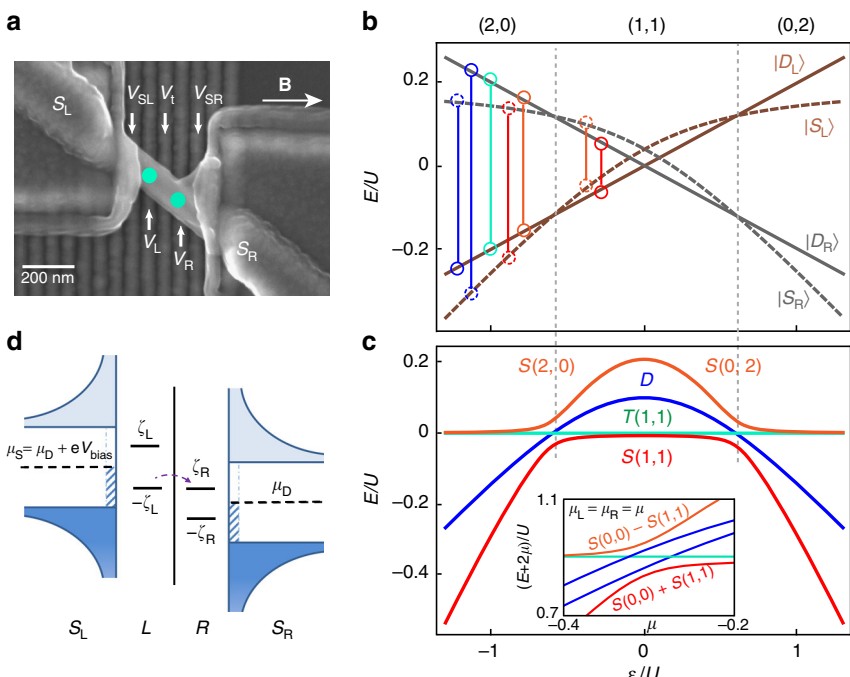

**Fig. 1** Superconducting double dot and its energy levels. **a** Scanning electron micrograph of the InSb nanowire device, *green circles* indicate positions of the two quantum dots. The two superconducting leads are $S_L$ and $S_R$. The double dots are defined and tuned by five local gates: $V_{SL}$, $V_L$, $V_t$, $V_R$, and $V_{SR}$. The direction of magnetic field **B** is indicated by arrow. **b** Spectrum of Andreev states in two quantum dots separated by a large barrier as a function of detuning $\epsilon$. On the *left* (*right*) dot, the ground states are $|S_{L(R)}\rangle$, $|D_{L(R)}\rangle$ and $|S_{L(R)}\rangle$ with dot occupations 2, 1, 0 (0, 1, 2), respectively. *Vertical lines* connect levels that hybridize to form molecular states plotted in **d**. **c** Molecular Andreev spectrum of a double quantum dot as a function of detuning (*main panel*) and energy level shift (*inset*). S(2,0) denotes singlet (2,0) configuration and similar for others. Charge configurations in **b**, **c** are labeled in **b** and separated by *dashed lines*. **d** Transport resonance at positive bias occurs when Andreev chemical potentials $-\zeta_L(V_L, \mu_S)$, and $\zeta_R(V_R, \mu_D)$ are aligned and bias ($V_{bias}$) compensates relaxation energies. The *hashed bars* depict subgap states included in the numerical model

the electrostatic energies on the two dots are changed in the opposite directions. From negative to positive detuning, the left dot is occupied with 2, 1, and 0 electrons, while the right dot is occupied with 0, 1, and 2. In the (2,0) and (0,2) double dot configurations, singlet states on both dots are lower in energy than doublets. In the (1,1) configuration, doublets are the ground states. We note that in the experimental system the charge occupations of the dot are unknown but the discussion is provided in terms of 0, 1, 2 electrons on each dot for clarity and because in superconducting systems properties most strongly depend on parity (even or odd) of the charge occupations.

When the two dots are tunnel-coupled, each of the states on one dot will hybridize with each of the states on the other dot (Fig. 1c). The new singlet states are $S(0,2)$, $S(2,0)$, and $S(1,1)$: these three states hybridize at their degeneracy points due to tunnel coupling. The four doublet states hybridized of $D(0,1)$ and $D(1,0)$, $D(2,1)$ and $D(1,2)$ are nearly degenerate at zero field and are designated as $D$ in Fig. 1c and are always the excited states. When the chemical potentials $\mu_L$ and $\mu_R$ on the left and right dots are tuned along the energy shift axis, such that $\mu_L = \mu_R = \mu$ the double dot can transition from (0,0) to (1,1) configuration. In this case, $S(0,0)$ and $S(1,1)$ are hybridized by superconducting correlations (Fig. 1c, inset). A new type of levels appears below the gap in a double quantum dot: the three triplet states $T_+(1,1) = (\uparrow,\uparrow)$, $T_-(1,1) = (\downarrow,\downarrow)$ and $T_0(1,1) = (\uparrow,\downarrow) + (\downarrow,\uparrow)$ trace back to the symmetric combinations of single dot doublet states. $T(0,2)$ and $T(2,0)$ are above the induced gap due to the large orbital energy and thus they do not correspond to bound Andreev states.

In experiment, source-drain voltage bias $V_{bias}$ is applied between $S_L$ and $S_R$ to tune the chemical potentials in the source and drain superconductors $\mu_S$ and $\mu_D$ (Fig. 1d and see more in Supplementary Fig. 12). On the left and right quantum dots, chemical potentials that correspond to transitions between

ground and excited Andreev bound states, $\pm\zeta_L$ and $\pm\zeta_R$, are arranged symmetrically around the chemical potential of the left (right) superconductor. The splitting between particle-like and hole-like Andreev resonances $+\zeta$ and $-\zeta$ on each dot is tunable with gate voltages on that dot. A resonance in conductance through the double dot occurs when $\mu_S - \mu_D = \zeta_L + \zeta_R$, and thus for each setting of gates $V_L$ and $V_R$ the transport resonance corresponds to a unique value of $|V_{bias}|$.

**Measurements.** Measurements below are focused on a double dot stability diagram presented in Fig. 2a (see Supplementary Fig. 1 for an expanded diagram). Four degeneracy points are observed at which the current has a local maximum. The upper-left maximum of current is lower than the other three. In reverse $V_{bias}$, the lower-right maximum has the lowest current. This is due to spin blockade which occurs between (1,1) and (0,2), or (2,0) double dot states due to Pauli exclusion (see Supplementary Figs. 2 and 3 for further evidence)[28]. Spin blockade is a manifestation of hybridized quantum states on the two dots, and it allows us to identify and label the parity of nine configurations in Fig. 2a. The regime is closely reproduced by a numerical model of the superconducting double dot discussed below, including the spin blockade regime (Fig. 2b). In differential conductance the double dot stability diagram is defined by arc-shaped resonances that connect the degeneracy points (Fig. 2c–f).

The arcs in double dot stability diagrams originate from loop-like resonances in gate vs. bias scans (Fig. 3 and see more in Supplementary Figs. 4 and 13). The loop resonances appear most clearly when one dot is fixed at a degeneracy point and the other dot is swept (Fig. 3a, b). Loop-like resonances are also observed when the energy levels on the two dots are tuned simultaneously (Fig. 3c-f), though deep within the (1,1) region the interdot tunnel

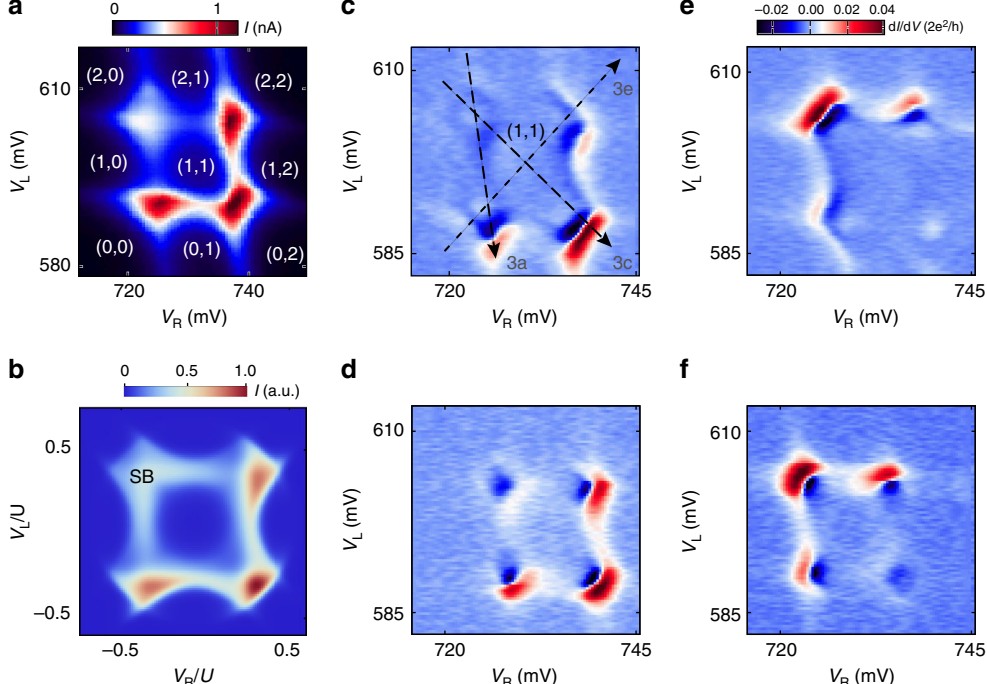

**Fig. 2** Stability diagrams. **a** Current at $V_{bias}= 200\,\mu V$ as a function of left and right gate voltages ($V_L$ and $V_R$). Parities of *double dot* configurations are indicated in brackets. **b** Numerically computed current at low interdot tunneling as a function of chemical potentials of left and *right dots* over charging energy U. "SB" marks the corner with numerically reproduced spin blockade. **c** Differential conductance d$I$/d$V$ at $V_{bias} = 200\,\mu V$ over the same gate voltage range as in **a**. *Dashed lines* indicate the cuts that correspond to panels in Fig. 3. **d–f** Differential conductance d$I$/d$V$ at $V_{bias} = -200, 50, -50\,\mu V$ over the same gate voltage range as in **a**

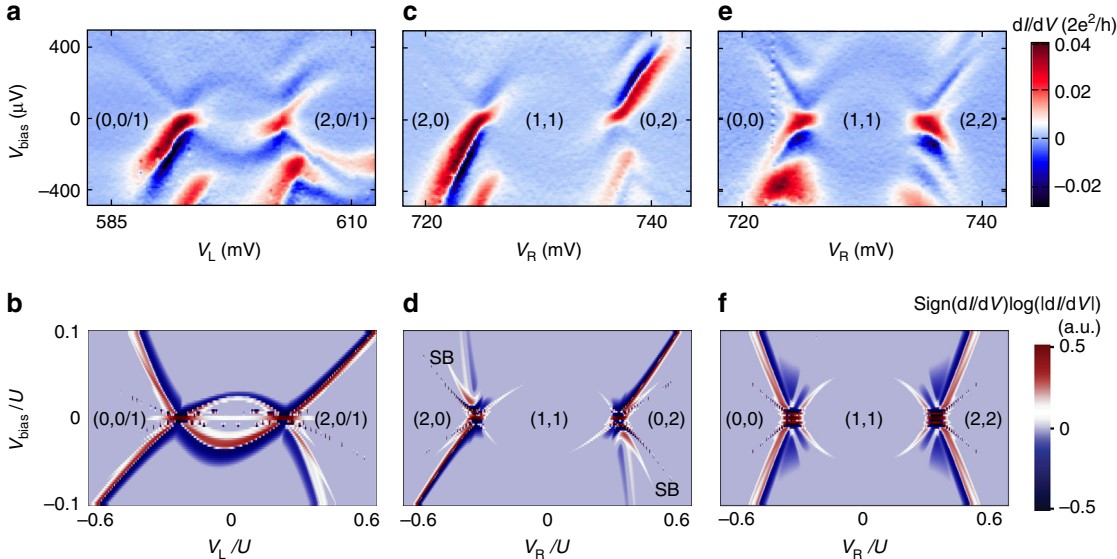

**Fig. 3** Bias spectroscopy of Andreev molecular states. **a**, **c**, **e** Source-drain bias spectroscopy along various cuts depicted by the *dashed-line arrows* in Fig. 2c, i.e., *right dot* is fixed to 0/1 degeneracy point and *left dot* is swept in **a**, along detuning in **c**, and along energy level shift axis in **e** (see additional complementary data in the Supplementary Fig. 4). Both $V_L$ and $V_R$ are tuned in each panel, but either $V_R$ or $V_L$ is used to denote the x-axis. Parity configurations indicated in brackets in each region. **b**, **d**, and **f**, Corresponding numerically computed differential conductance as function of *left dot* chemical potential with fixed right dot chemical potential **b**, along detuning **d** and energy level shift axis **f**. "SB"s in **d** mark the numerically reproduced spin blockade corners at positive and negative biases

coupling is reduced and the current is suppressed. The loop resonances are accompanied by copies in negative differential conductance. This is because on resonance (Fig. 1d) current has a maximum, hence differential conductance changes from positive to negative. The origin of arcs in Fig. 2 can now be understood: indeed, if Andreev resonances are loop-like in $V_{bias}$ for any cut through the double dot stability diagram, a scan at fixed $V_{bias}$ would reveal arc resonances when $V_{bias}$ matches the interdot Andreev resonance condition.

The observed Andreev loops are closed, i.e., the conductance resonances reach zero bias. This is counter-intuitive given that both leads of the system are superconductors and thus an energy gap is expected around zero bias (see Supplementary Fig. 11 for bias spectroscopy plots of single dots)[13, 15]. We ascribe this to subgap quasi-particles that enable single-particle transport through the Andreev molecular states. When this effect is included in the numerical model, simulations reproduce the closed loops and negative differential conductance, as well as bias asymmetries due to spin blockade (Fig. 3b, d, f). We model each lead as being composed of two parts: a conventional super-conductor with a hard superconducting gap and a normal Fermi gas with gapless excitations. The electrochemical potentials of the normal and the superconducting parts are pinned together at the value set by the voltage applied to the physical lead. In our model, Andreev reflection off the superconducting part results in the formation of Andreev molecules. The normal part induces transitions between the Andreev molecular states (see Supplementary Note 2 for details). For simplicity, the model assumes leads with a superconducting gap much larger than the single dot energy $U$[29].

We investigate the spin structure of Andreev molecular states by monitoring the evolution of subgap transport features in magnetic field. In Fig. 4a we plot differential conductance as a function of magnetic field and source-drain bias for a double quantum dot in the (2,2) configuration. At zero magnetic field, we observe two peaks, one at positive bias and one at negative bias. The application of magnetic field results in the splitting of both

peaks. Two of the peaks move to higher bias toward the gap edge, while the other pair meets at zero bias. The two merged resonances stick to zero bias at finite field. This effect has been investigated as a signature of Majorana fermions[20]. Here, given the narrow range of field over which the zero bias peak is observed, we associate it with level repulsion from the gap edge or from other subgap states[18]. By comparing measurements to numerical spectra and transport calculations, we assign the peaks to the transitions between the $S(2,2)$ ground state and the $D(\uparrow,2)$ and the $D(\downarrow,2)$ excited states (Fig. 4b, c). Magneto-transport of the double quantum dot system in the (0,0), (0,2), and (2,0) configurations is qualitatively the same as in the (2,2) configuration (see more Supplementary Figs. 5–7 and 14 and measurements in strong coupling regimes in Supplementary Figs. 8–10).

In the (1,1) configuration only a single pair of differential conductance peaks is observed at all fields, one at positive and one at negative bias (Fig. 4d). Both peaks shift to higher bias at higher magnetic fields. The explanation for this behavior originates in the Andreev molecular level structure depicted in Fig. 4e. The low energy manifold consists of $S(1,1)$ ground state that is almost degenerate with the three triplet states $T_+$, $T_0$, $T_-$. At finite field $T_+$ plunges below the $S(1,1)$ and becomes the ground state. Transitions from this triplet state are allowed only to the doublet states $D(\uparrow,0)$, while transitions to $D(\downarrow,0)$ are strongly suppressed because they involve an additional spin flip. Both states $T_+$ and $D(\uparrow,0)$ shift to lower energies with magnetic field, but the triplet states shifts with $g\mu_B B$ while the doublet states shifts with $g\mu_B B/2$, thus the energy difference between them grows with field. Transport calculations using our detailed model confirm this picture (Fig. 4f).

Odd total parity configurations (0,1), (1,0), (2,1) and (1,2) offer a richer variety of transport behavior (Fig. 4g, and Supplementary Figs. 5 and 6). The common features include asymmetry with respect to bias and kinks in the conductance peaks at which the effective g-factor increases. In some regimes we also observe the magnetic field induced splitting of conductance peaks into as many as three sub-peaks. In Fig. 4h

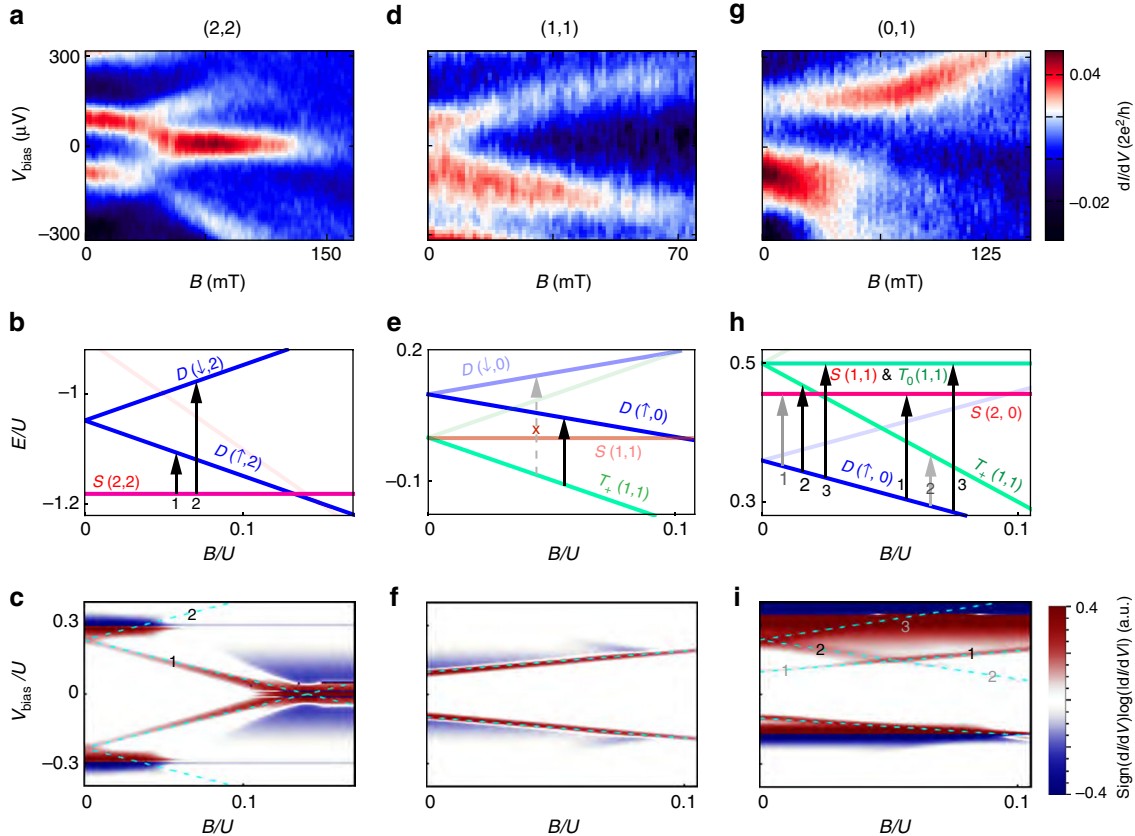

**Fig. 4** Magnetic field evolution of Andreev molecular states. **a**, **d**, **g** Bias spectroscopy of Andreev resonances as a function of magnetic field for (2,2), (1,1) and (1,0) double dot configurations. S(2,2) denotes singlet (2,2) configuration. **b**, **e**, **h** Numerically computed spectra of Andreev molecular states as a function magnetic field for $V_{bias} = 0$. The *black (gray) arrows* and numbers label the allowed transitions in the simulated spectra (**b**, **e**, **h**) and the associated high (*low*) conductance resonances in the numerical $dI/dV$ transport plots (**c**, **f**, **i**). In **e** the crossed dashed arrow labels the forbidden transition between $T_+(1,1)$ and $D(\downarrow,0)$. The *light blue* dashed lines in **c**, **f**, **i** plot the bias voltage at which the levels on the dots come into resonance. The $dI/dV$ plots use the same model parameters as in the spectrum plots

we plot the Andreev molecular spectrum in the (0,1) configuration as a function of magnetic field at zero bias. While $D(0,\uparrow)$ is the well-separated ground state at finite field, there are two singlet states ($S(0,2)$ and $S(1,1)$) and two triplet states ($T_+$ and $T_0$) that can contribute to transport (transport via the state $T_-$ requires a spin flip and is therefore suppressed). Numerically computed transport demonstrating both a kink feature as well as the asymmetry with respect to bias, is plotted in Fig. 4i. The model indicates that the origin of the kink feature is that as $B$ increases the $D(\uparrow,0) \to T_+(1,1)$ transition (labeled 2 in Fig. 4h, i) becomes dimmer while the $D(\uparrow,0) \to S(2,0)$ transition (labeled 1 in Fig. 4h, i) becomes brighter. In the model, the dimming and brightening of the transitions is associated with proximity to the interdot resonances that occur at higher bias. The bias asymmetry is associated with the different parities of the left and right dots, the asymmetry flips if the parities are switched (Supplementary Note 2).

## Discussion

The elucidation of Andreev molecular spectra and of their evolution in magnetic field opens several avenues for future research. Andreev molecule is a building block for the emulation of the Kitaev chain model[9, 10], in which tuning of longer quantum dot chains is to be performed pairwise along the chain. This taps into the largely unexplored potential of semiconductor systems for quantum simulation research[30]. Simulations of quantum dynamics of Andreev states in quantum dots chains can be

attempted in hard gap nanowires[26, 31, 32]. In topological qubits, double quantum dots have been proposed for fusion and readout of Majorana quantum states[22]. These operations transmute topologically protected Majorana states into Andreev molecular states. Andreev molecules with topologically superconducting reservoirs will become building blocks of topological quantum circuits, and can be realized in the same nanowires with longer quantum dots (200 nm or longer) subjected to higher magnetic fields (0.5 Tesla)[26, 33].

## Methods

**Fabrication.** The nanowires (diameter 100 nm) are grown in the 111 crystal orientation by metalorganic vapor phase epitaxy from gold catalysts, as described in ref. [12]. Local gate electrodes (pitch 60 nm) are defined by electron beam lithography and electron beam evaporation of Ti(5 nm)/Au(10 nm) on thermal silicon oxide. The gate electrodes are then covered by atomic layer deposition (ALD) grown HfO$_2$ (10 nm). Single InSb nanowires are transferred by a micro-manipulator. The superconducting contacts are Ti/NbTi/NbTiN (5/5/150 nm). Prior to sputtering the nanowires are passivated in ammonium sulfide to remove the native oxide.

**Measurement techniques.** The measurements are performed at 35 mK in a dilution refrigerator. A d.c. voltage bias is applied to the left superconducting lead ($S_L$) and the current from the right superconducting lead ($S_R$) to the ground is measured by a current amplifier. To measure the differential conductance, a standard lock-in technique is used (77 Hz, 5 μV).

**Data availability.** The data sets generated during and/or analyzed during the current study are available in the 4TU data center repository, http://data.4tu.nl/repository/uuid:e99d1ab7-2e82-447e-a314-f230a0da4a95.

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

## Acknowledgements

We thank R. Aguado, A. Akhmerov, S. De Franceschi, E. Lee, V. Liu for valuable discussions. Work is supported by Charles E. Kaufman Foundation (S.M.F. and D.P.), NSF DMR-125296, ONR N00014-16-1-2270 (S.M.F.) and AFOSR FA9550-12-1-0057 (A.D.).

## Author contributions

D.C., S.R.P., and E.P.A.M.B. grew InSb nanowires. Z.S. and M.H. fabricated devices. Z.S. and S.M.F. performed the measurements and analyzed data. A.B.T., A.J.D., and D.P. performed numerical simulations. All authors wrote the manuscript.

## Additional information

**Competing interests:** The authors declare no competing financial interests.

