## [Peer Review File · Nature Communications]

Reviewer #1 (Remarks to the Author):

The paper presents transport data on a double quantum dot device with two superconducting leads. The transport spectroscopy shows signatures of the different double-dot charge configuration which can be tuned by gates.

The data are compared to numerical simulation based on master equations with single particle tunneling between the dots and the leads. In order to fit the data, the authors assume the superconducting leads to have a subgap density of states. Overall, the theoretical simulations fail to reproduce many of the features in the data. Some of the observed lines are reproduced, for example in the Fig. 3., but the main feature in the (1,1) region has the wrong sign (the simulation is plotted on a log scale and the data on a linear scale, which makes them hard to compare. Also I assume what is really plotted is $\text{sign}(dI/dV)\log|dI/dV|$?)

The motivation for the paper is presented as the prospects for using Andreev dots for quantum simulation devices. However, the paper does not even address any dynamical aspects of the devices or mention how one should operate the device. Moreover, the lack of a gap in the spectrum of the leads seems to forbid such dynamical operation, because of the presumably very short resulting life times. This is also not discussed in the paper.

The physics of molecular double dot states is by now well-known physics and it is indeed an interesting topic to investigate how this carries over to a setup with superconducting leads. However, in my opinion, this paper does not sufficiently advance the field of quantum dots coupled to superconductors to warrant publication in Nature Communications. The amount of data is rather small. A potentially new and interesting topic could be a systematic study of what happens in the strong dot-lead coupling limit, where one could expect an interplay of Shiba states formed in the leads and the molecular double-dot states. The present paper does indeed focus on this strong dot-lead coupling case, and therefore the Shiba states might play a role and potentially be a reason for the lack of agreement with the theoretical model presented in the paper. In my opinion, the paper should have analyzed this aspect more and discussed its importance for the discrepancy between experiment and the lowest-order tunneling model.

Reviewer #2 (Remarks to the Author):

This article describes experiments on a double quantum dot tunnel coupled to superconducting leads. Quantum dots coupled to superconductors are of great interest because they may host Majorana fermions. A proximity-coupled double quantum dot is in principle sufficient to test some aspects of topological quantum computation with Majoranas.

In this work, the quantum dots are formed in an InSb nanowire and are proximity coupled to NbTiN superconducting leads. Andreev bound states form in the double quantum dot and modify its level structure. This paper investigates the bias and magnetic field dependence of transport through these states, finding good agreement with a theoretical model. Andreev bound states are typically probed with a normal metal contact, which has a non-vanishing density of states at the Fermi energy. Here, however, quasiparticles in the superconductor, which does not have a hard gap, fulfill this purpose.

The experimental work is high quality, and the experimental platform is interesting. I support publication after the following points have been addressed.

1. My general comment is that the manuscript would be improved with more context for the work overall and each of the sections individually. For example:

a. What is the big-picture importance of these results? The measurements agree with the model, but what have we learned that will help us produce or braid Majoranas? What are the next steps for testing fusion rules in this platform?

b. In the first paragraph on p. 3, the occupation number of quantum dots is discussed. However, the stability diagram in the supplement shows more than 2 electrons in at least one dot. I was confused until line 78 on p. 5, where it is mentioned in passing that "the dot occupations are higher than their parities." It would be helpful to discuss this earlier and more explicitly.

c. An intuitive discussion in the manuscript of the mechanism for the current in the arcs in Fig. 2 would be helpful. In the manuscript, this is broadly described as a resonance between states. In a normal double quantum dot, for example, current at the triple points results from resonance between the leads and states in each of the dots, and current along lines connecting the triple points results from cotunneling. Is an analog of cotunneling the mechanism behind the current in the arcs away from the "triple points"? There is a substantial calculation in the supplement for the current through the double dot, and an intuitive summary of the main idea would be useful in the manuscript, especially since this is the essential measurement for the paper. A schematic figure including a ladder of dot states, perhaps similar to Fig. S14, might also be useful in the main text.

2. Does the "amount" of normal Fermi gas in the leads match the measured lead resistance, or is this a free parameter in the model?

3. What is the critical temperature and field of the superconducting leads? It would be interesting to see the normal-superconducting crossover in the charge stability diagram if the leads can be made normal. I understand that this may not be possible with NbTiN, however.

In summary, this manuscript explores an interesting physical system of a double quantum dot coupled to superconducting leads, which may have importance for topological quantum computing. The observed results in this paper are well described by a theoretical calculation. I support publication once the above points have been addressed.

REPLY TO REFEREES:

We thank the referees for evaluating the Andreev Molecule manuscript and providing their comments. We would like to submit a new version of the manuscript that has been re-worked based on the referee feedback. We are looking forward to the new round of review and provide detailed replies below (in bold font).

Reviewer #1 (Remarks to the Author):

The paper presents transport data on a double quantum dot device with two superconducting leads. The transport spectroscopy shows signatures of the different double-dot charge configuration which can be tuned by gates.

The data are compared to numerical simulation based on master equations with single particle tunneling between the dots and the leads. In order to fit the data, the authors assume the superconducting leads to have a subgap density of states. Overall, the theoretical simulations fail to reproduce many of the features in the data. Some of the observed lines are reproduced, for example in the Fig. 3., but the main feature in the (1,1) region has the wrong sign (the simulation is plotted on a log scale and the data on a linear scale, which makes them hard to compare).

We begin by emphasizing that our model incorporates both single particle tunneling AND Andreev reflection. Andreev reflection can be thought of as pair tunneling and results in formation of Andreev bound states on the quantum dots. We also comment that in our view Andreev bound states on a quantum dot with odd occupancy are equivalent to Shiba states. The difference being the origin of the magnetic impurity: Shiba states are generated by a magnetic atom while Andreev bound states are generated by the quantum dot with repulsive interactions.

We chose the basic model of coupled Andreev Bound States that neglects some aspects of the realistic system for two reasons: first, it captures the fundamentals of Andreev molecules -- their coherent charge and spin structure. Thus, the model has significant pedagogical value for a broad audience. Second, when the subgap density of states is added, the model reproduces key features of the experiment, especially the lowest energy Andreev resonances (see below for a detailed discussion of the features that the model reproduces with side-by-side figure comparison).

We have re-done the simulations throughout the paper to improve comparison with experiment further. In particular, Figure 3 has been changed to a new layout in which each experimental panel is accompanied by a theoretical panel directly below the experimental one.

We first discuss Figure 3 as it has been highlighted by the referee. Both the model and the experiment show that each resonance consists of a positive conductance peak followed by a negative conductance dip (see Fig. R1 for detailed comparison, also Fig. R2 and Fig.R3), this is because current through the device spikes on resonance, and a derivative of that produces two peaks of opposite sign. In experiment, the height is not the same for positive and negative conductance peaks. This may perhaps create the impression that the simulated feature has “the wrong sign” - or “blue” color vs. “red”. This is due to a combination of the exact energy dependence of the density of states as well as the energy dependence of inelastic processes in the double dot. By adjusting the fit parameters we are able to emphasize the “red” or the “blue” part of the resonance.

To the authors, the main feature in Figure 3 are the loop-like resonances, which are reproduced by the model. This can be seen in Fig.R1. For other plots, they are depicted by the green dashed arrows (1 and 2) in Fig. R2 and by the green dashed arrows (2 and 3) in Fig.R3. Deep inside the (1,1) region the visibility of the loop resonances is low both in theory and experiment due to reduced tunneling. The symmetry with respect to bias in Fig.R1 and the anti-symmetry in Fig.R2 are present in both experimental and theoretical plots. The weaker resonance near zero bias on the experimental plot in Fig.R3 due to the degeneracy of the other dot is reproduced by theoretical plot on the right.

Moving to Figure 4, the direction in which Andreev resonances shift in bias when magnetic field is

applied is correctly reproduced by the model (Figure 4 d,e,f in the main text). This could be considered another main feature of the (1,1) regime since it establishes the correct spin structure of the Andreev molecular eigenstates in the (1,1) region. The agreement also spreads to the (even, even) configuration (Figure 4 a,b,c) and to the (even, odd) configuration (Figure 4 g, h, i).

Similarly, in Figure 2 the main feature of the data is the concave-arcs connecting four transport-allowed areas rather than honeycomb shape of the (1,1) region. Both experimental panel a and theoretical panel b show resonant current at four corners connected by the arcs. The upper-left corner and the two connecting arcs have reduced current due to spin blockade (See Fig.R4). In the resubmitted version, we have included additional data in Figure 2 obtained at different biases and for different bias directions to fully elucidate the concave nature of the Andreev arcs connecting the parity degeneracy points.

While we argue above that the basic model reproduces the key characteristics of Andreev molecules observed in the experiment, the referee is right that not all features of the experiment are reproduced by this basic model. We ascribe the difference in the features between the experimental data and the theoretical model to a number of additional processes that we did not model. These processes include: non-monotonic density of states of single particle excitations in the leads, inelastic processes in which electrons exchange energy with the phonon bath, and spin-orbit coupling.

Motivated by the reviewers comments, we have developed a complementary rate model which takes into account inelastic processes, and spin-orbit coupling at the cost of inter-dot coherence (see Figure R5 below). The inelastic rate as a function of energy is a free function that can be adjusted arbitrarily to improve match to experiment. The processes included in the complementary model were not found to alter the conclusions qualitatively. While some of the features e.g. linewidths of the resonances, are better reproduced by the complementary model, others such as sharp negative differential conductance features, are lost, indicating the importance of inter-dot coherence in the experiment. To maintain pedagogical value of the manuscript, we keep results from the basic model in the manuscript but we share the results from the complementary model with the reviewers as figure R5.

Another important type of features found in the experiment but not in the model are extra resonances at higher bias (e.g. Figure R2 below feature labeled 4 and 4'). The nanowire segments under the superconducting leads are mesoscopic objects and the density of states in them is not smooth giving rise to conductance resonances and consequently extra features in the data. These features are currently being analyzed with a different group of theory colleagues and will be addressed in a separate manuscript. The present manuscript contains a comprehensive study of the lowest-bias features.

We indeed use non-identical color scales for theory and experiment to clearly separate experiment and theory. To allow for a detailed comparison between experiment and theory, we provide digital raw data of all experimental and theoretical plots in the manuscript, main text, supplementary and Fig.S5 in a zipped file. The instruction to plot them is also attached.

Also I assume what is really plotted is $\text{sign}(dI/dV)\log|dI/dV|$?)

The referee is right, we thank the referee for noticing this and we have corrected the labels in the theory plots.

The motivation for the paper is presented as the prospects for using Andreev dots for quantum simulation devices. However, the paper does not even address any dynamical aspects of the devices or mention how one should operate the device. Moreover, the lack of a gap in the spectrum of the leads seems to forbid such dynamical operation, because of the presumably very short resulting life times. This is also not discussed in the paper.

The referee is correct that without a hard superconducting gap the prospects for simulating quantum dynamics with our superconductor/quantum dot system are bleak. However, as hard gap has been demonstrated with other superconductors, such as Al, and even with NbTiN, this does not present a

fundamental obstacle. However, as was stated in the abstract we plan to use these dots to investigate the ground states of quantum many-body Hamiltonians, which is an important type of quantum simulation. In the case of Andreev dots, we are interested in the ground state of a Kitaev chain, which generates Majorana fermions. Our setup is a powerful way to study topological superconductivity and is within the reach of this experimental system, since it is already extensively used in Majorana experiments. The sense in which our work is useful for quantum simulation is now additionally discussed in the conclusion paragraph.

The physics of molecular double dot states is by now well-known physics and it is indeed an interesting topic to investigate how this carries over to a setup with superconducting leads. However, in my opinion, this paper does not sufficiently advance the field of quantum dots coupled to superconductors to warrant publication in Nature Communications.

On the experimental side, this work does not follow any previous work in any incremental fashion. There are no previous direct experimental studies of the full spectroscopy and spin structure of Andreev molecular states created in fully-tunable devices reported in the literature. Accidental double quantum dots coupled to superconductors have been made and investigated to a degree (see the first reference below), and quantum dots that are not tunnel-coupled have been studied in other contexts such as a Cooper pair splitter (see the second reference below).

J-D. Pillet et al, Andreev bound states in supercurrent-carrying carbon nanotubes revealed, Nature Physics, 6, 965-969 (2010)

L. Hofstetter et al., Cooper pair splitter realized in a two-quantum-dot Y-junction, Nature, 461, 960-963 (2009)

The amount of data is rather small.

In the resubmitted paper, we have added 3 data sets from a new generation device and 1 simulation plot for it (4 simulation plots generated by the complementary model are presented in this letter). Thus we now present in total 76 distinct experimental and 35 distinct theoretical data sets (excluding schematic plots), including 24 in the main text and 86 in the supplementary information. We hope that the referee finds the amount of data to be sufficient.

A potentially new and interesting topic could be a systematic study of what happens in the strong dot-lead coupling limit, where one could expect an interplay of Shiba states formed in the leads and the molecular double-dot states. The present paper does indeed focus on this strong dot-lead coupling case, and therefore the Shiba states might play a role and potentially be a reason for the lack of agreement with the theoretical model presented in the paper. In my opinion, the paper should have analyzed this aspect more and discussed its importance for the discrepancy between experiment and the lowest-order tunneling model.

In the strong dot-lead coupling regime, the ground state Andreev level in (1,1) or any odd occupation on either of the dots is always spin-singlet state due to BCS and Kondo screening. We now include data from a new device in which such strong coupling limit has been realized (See Fig.S10 in the supplementary information). The key qualitative difference between the weak coupling regime and this strong coupling regime is that the resonances in the bias-gate space become anti-crossing-like rather than loop-like. This behavior is also reproduced by the model. With the addition of this strong coupling regime we now cover the exhaustive range of double dot tuning regimes for this system.

There are no Shiba states in the leads, neither in the superconductor which has no magnetic impurities, nor in the segments of the nanowire adjacent to the double dot. When the double dot is removed by gates, the two terminal conductance through the nanowire shows no quantum dots and the two-terminal resistance is 4 k Ω indicating highly transmitting regime. Thus there are no localized spins in the nanowire segments outside of the two quantum dots.

Fig.R1 Comparison of the signs of experimental and simulated resonances. (a) The experimental measurement of the subgap resonance when chemical potentials of both dots are increased. The line-cut, depicted by the red dashed line in (1,1), is plotted in (b) where a positive dI/dV peak followed by a negative dI/dV dip is shown at either bias. (c) The associated simulated plot. The line-cut at a similar position, depicted by the black dashed line in (1,1) is plotted in (d) in linear scale where positive peaks followed negative dips. Note that the color scales of (a) and (c) set white to be positive values.

Fig.R2 Comparison of the features of experimental and simulated plots along detuning. Green dashed line arrows 1 show the resonances in (2,0) on both experimental and theoretical plots, and the bias anti-symmetry in this regime (more current at negative bias in (2,0) and more current for positive bias in (0,2) due to spin blockade). Arrows 2 show the resonances in (1,1) on both experimental and theoretical plots. Red number 3 depicts the regimes where the resonances are weak due to reduced interdot tunnel coupling. Features at higher bias depicted by arrow 4 and 4' are not well-

reproduced and are a subject of a subsequent study with different co-authors.

Fig.R3 Comparison of the features of experimental and simulated plots when chemical potential of one dot is fixed. The experimental measurement and simulation of the subgap resonances along chemical potential of left dot solely. Green dashed line arrows 1 depict the weaker resonances near zero bias on both experimental and theoretical plots. Arrows 2 and 3 show that the loop-like resonances consists of positive then negative dI/dV peaks on both experimental and theoretical plots.

Fig.R4 Comparison of experimental and simulated stability diagrams. Green dashed line arrow 1 depicts the weaker corner due to spin blockade on both experimental and theoretical plots. Arrows 2 shows the arcs connecting the four corners. Arrow 3 shows that the opposite corner has the highest strength. Notice that the arcs connecting the upper-left corner are weaker, which is reproduced by the simulation as well.

Fig.R5 Simulations generated by the complementary model that considers inelastic tunneling and spin-orbit coupling. **a, b, c** display the subgap resonances along detuning, simultaneous tuning of chemical potentials on both dots, and along chemical potential of left dot while the chemical potential of right dot is fixed when the superconductor-quantum dot coupling is not very strong. They correspond to Fig.3d,f,b in the main text where the basic model is used. **a, b** and **c** share the color upper color bar. **d** displays the simulated subgap resonances in the strong superconductor-quantum dot coupling regime when the right dot chemical potential is fixed and the left dot chemical potential is swept. Notice that the resonance is anti-crossing-like in **d** while the resonance is crossing-like in **c**.

Reviewer #2 (Remarks to the Author):

This article describes experiments on a double quantum dot tunnel coupled to superconducting leads. Quantum dots coupled to superconductors are of great interest because they may host Majorana fermions. A proximity-coupled double quantum dot is in principle sufficient to test some aspects of topological quantum computation with Majoranas.

In this work, the quantum dots are formed in an InSb nanowire and are proximity coupled to NbTiN superconducting leads. Andreev bound states form in the double quantum dot and modify its level structure. This paper investigates the bias and magnetic field dependence of transport through these states, finding good agreement with a theoretical model. Andreev bound states are typically probed with a normal metal contact, which has a non-vanishing density of states at the Fermi energy. Here, however, quasiparticles in the superconductor, which does not have a hard gap, fulfill this purpose.

The experimental work is high quality, and the experimental platform is interesting. I support publication after the following points have been addressed.

1. My general comment is that the manuscript would be improved with more context for the work overall and each of the sections individually. For example:

a. What is the big-picture importance of these results? The measurements agree with the model, but what have we learned that will help us produce or braid Majoranas? What are the next steps for testing fusion rules in this platform?

We have added a discussion of the implications of this work for quantum simulation research, and for topological quantum computing in the concluding paragraph. Hybridization of Andreev states shares many features with the hybridization (or fusion) of Majorana states and thus we expect similar effects to manifest in fusion or braiding devices. In order to perform Majorana fusion, for example, molecular states should be studied in longer double quantum dots, with the length exceeding the Majorana length (100 nm in InSb/NbTiN devices), and at high enough magnetic fields (order 0.5 Tesla).

b. In the first paragraph on p. 3, the occupation number of quantum dots is discussed. However, the stability diagram in the supplement shows more than 2 electrons in at least one dot. I was confused until line 78 on p. 5, where it is mentioned in passing that "the dot occupations are higher than their parities." It would be helpful to discuss this earlier and more explicitly.

The paper has been changed to indicate early on that the quantum dot occupations are higher than the parities indicated in figures and throughout the paper.

c. An intuitive discussion in the manuscript of the mechanism for the current in the arcs in Fig. 2 would be helpful. In the manuscript, this is broadly described as a resonance between states. In a normal double quantum dot, for example, current at the triple points results from resonance between the leads and states in each of the dots, and current along lines connecting the triple points results from cotunneling. Is an analog of cotunneling the mechanism behind the current in the arcs away from the "triple points"? There is a substantial calculation in the supplement for the current through the double dot, and an intuitive summary of the main idea would be useful in the manuscript, especially since this is the essential measurement for the paper. A schematic figure including a ladder of dot states, perhaps similar to Fig. S14, might also be useful in the main text.

We have added a diagram illustrating the resonance process to Figure 1 and discussed the mechanism based on these diagrams in the main text. We thank the referee for this suggestion which greatly improves the clarity of the manuscript.

2. Does the "amount" of normal Fermi gas in the leads match the measured lead resistance, or is this a free parameter in the model?

This is a free parameter in the model which cannot be determined independently from other parameters based on, e.g. lead resistance. Qualitatively the features look the same no matter which normal density of states is used.

3. What is the critical temperature and field of the superconducting leads? It would be interesting to see the normal-superconducting crossover in the charge stability diagram if the leads can be made normal. I understand that this may not be possible with NbTiN, however.

The critical temperature of NbTiN is 14K while the critical field exceeds 10 Tesla, thus it is not feasible to observe a transition into a non-superconducting state in the same device. Previously published works demonstrate normal double dots in InSb nanowires and can be used for comparison. For example, spin blockade reported in reference (S. Nadj-Perge et al., *Spectroscopy of Spin-Orbit Quantum Bits in Indium Antimonide Nanowires*, PRL, 108, 166801 (2012)) is in good agreement with the observations here.

In summary, this manuscript explores an interesting physical system of a double quantum dot coupled to superconducting leads, which may have importance for topological quantum computing. The observed results in this paper are well described by a theoretical calculation. I support publication once the above points have been addressed.

Reviewer #2 (Remarks to the Author):

I believe the authors have addressed the comments of the reviewers sufficiently to warrant publication. To summarize my thoughts:

1. The experimental platform is interesting and relevant, especially for topological quantum computing.
2. The main features in the data are reproduced by the authors' theoretical model. As the first reviewer points out, and as the authors agree, the data and predictions do not agree exactly in every case, but I think the overall agreement is good enough to say that we understand the main features of the data.
3. The expanded discussion about the mechanism of the current through the double dot and the context given in the concluding paragraph for the work as a whole improve the readability of the manuscript.